# Maternal Exposure to Potentially Toxic Metals and Birth Weight: Preliminary Results from the DSAN-12M Birth Cohort in the Recôncavo Baiano, Brazil

**DOI:** 10.3390/ijerph20136211

**Published:** 2023-06-23

**Authors:** Homègnon A. Ferréol Bah, Nathália R. dos Santos, Erival A. Gomes Junior, Daisy O. Costa, Victor O. Martinez, Elis Macêdo Pires, João V. Araújo Santana, Filipe da Silva Cerqueira, José A. Menezes-Filho

**Affiliations:** 1Institute of Collective Health, Federal University of Bahia, Salvador 40110-040, Brazil; 2Laboratory of Toxicology, College of Pharmacy, Federal University of Bahia, Salvador 40170-115, Brazil; 3Graduate Program in Pharmacy, College of Pharmacy, Federal University of Bahia, Salvador 40170-115, Brazil; 4Graduate Program in Food Science, College of Pharmacy, Federal University of Bahia, Salvador 40170-115, Brazil

**Keywords:** potentially toxic metals (PTM), birth weight, fetal growth, sex-specific effect of PTMs

## Abstract

Prenatal exposure to potentially toxic metals (PTM) may impair fetal growth (FG). We investigated the relationship between maternal exposure to lead (Pb), cadmium (Cd) and manganese (Mn) and birth weight (BW) of 74 newborns. Blood was collected during the second trimester of pregnancy to determine Pb (PbB) and Cd (CdB), while hair (MnH) and toenails (MnTn) were used for Mn. Samples were analyzed by graphite furnace atomic absorption spectrophotometry (GFAAS). Sociodemographic and BW data were collected from questionnaires and maternity records, respectively. The medians (P25th–P75th) of PbB, CdB, MnH, and MnTn were, respectively, 0.9 (0.5–1.8) µg/dL; 0.54 (0.1–0.8) µg/L; 0.18 (0.1–0.4) µg/g; and 0.65 (0.37–1.22) µg/g. The means (standard deviation) of birth weight according to sex were 3067 (426.3) and 3442 (431) grams, respectively, for girls and boys. MnTn presented an inverse correlation with the BW/gestational age ratio for girls (rho = −0.478; *p* = 0.018), suggesting the effect of sex modification. Although BW correlation with CdB was not statistically significant, hierarchical linear regression (beta = −2.08; 95% CI–4.58 to 0.41) suggested a fetotoxic effect. These results confirmed the threat PTMs may represent and the need for more extensive research to elucidate their role in inadequate FG in developing countries.

## 1. Introduction

Low birth weight (LBW), defined as birth weight less than 2500 g [1], represents a risk factor for neonatal morbidity and mortality [2]. It has been associated with dwarfism, developmental delays, and chronic conditions such as heart disease, obesity and diabetes [3,4]. Almost 15% of babies worldwide are born with LBW; this percentage is lower for the Latin America and Caribbean region (8.7%) [5]. In Brazil, between 2011 and 2018, the prevalence was 9.6% [3], which is lower than the level reported in some countries on other continents (Asia and Africa) but higher than the European prevalence (6.5%) [5]. According to Paixão et al. [3], children with LBW in Brazil have more than 25 times the likelihood of neonatal death compared to babies born in the healthy weight range.

As a preventable event, LBW highlights a severe issue of health inequity, and it is, therefore, urgent to investigate possible approaches to improving this situation [6]. LBW has multiple causes, and reducing it requires improving nutritional aspects, ensuring a favorable context for maternal and child health, strengthening social support [5], and reducing environmental contamination [7]. One cause of LBW that has been investigated is exposure to potentially toxic metals (PTMs) such as cadmium (Cd), lead (Pb), and manganese (Mn). Cd and Pb are xenobiotics without any physiological role for living beings and with proven toxicity in animals and epidemiological studies [2,8,9]. Mn is a micronutrient that can harm health at deficient or excessive levels [10]. There is also discussion in the literature about possible toxic effects of PTMs even at low exposure levels [11,12]. These PTMs are of great interest for maternal reproductive health and fetal well-being due to their ability to cross the placenta, fetal toxicity, ubiquity as a natural component of the environment, and environmental liabilities [13,14,15,16]. Evidence has already been presented in the literature on the negative impacts of high and low levels of exposure to PTMs on fetal growth (FG) parameters such as birth weight, head circumference, length, and chest circumference [2,17,18,19,20].

Humans are commonly exposed to different types of pollutants simultaneously. There is growing concern about the consequences of multiple exposures to PTMs. Indeed, some studies have already demonstrated the possibility of combined effects of PTMs as a mechanism of toxicity [21,22,23,24]. However, there is a lack of knowledge regarding this issue in low- and middle-income countries. We recently presented the magnitude of exposure to PTMs in pregnant women in two municipalities in the Recôncavo Baiano (Brazil) included in the DSAN-12M birth cohort [25]. In this exploratory study, we reported low levels of Mn and Pb. In contrast, exposure to Cd measured in the blood (CdB) was relatively high, similar to levels in populations living in contaminated sites and occupationally exposed individuals.

Given the above, this study aimed to (i) investigate the association between exposure to these metals and children’s birth weight; and (ii) investigate the possibility of a combined influence of these metals on birth weight, taking into account the social context of the study population.

## 2. Materials and Methods

### 2.1. Study Population

This study population derived from the project “Socioenvironmental Determinants of Child Neurodevelopment” (DSAN-12M), which examined a prospective birth cohort in the municipalities of Nazaré das Farinhas and Aratuípe in the Recôncavo Baiano region of Brazil. More detailed information is presented elsewhere [25]. 

*The recruitment* was based on the network of primary care units (PCU) of the National Health System (SUS is its acronym in Portuguese), including 11 units in Nazaré and 4 units in Aratuípe. During prenatal consultations, the pregnant women were invited to receive the research team at their homes for explanations about the project goals and methods. If they accepted, adult participants or legal guardians (of pregnant adolescents) signed the consent form, while minors were asked to sign an assent form. The enrollment of 164 participants was carried out in two phases due to the COVID-19 pandemic: from July 2019 to March 2020, and from July 2021 to September 2022. 

This prospective study considered pregnant women’s exposure from insertion until delivery and the outcome of the newborn’s weight. The maternity wards of Hospital Luís Argolo (in the nearby municipality of Santo Antônio de Jesus) and the Santa Casa da Misericórdia (in Nazaré das Farinhas) were included in the survey to collect information from medical records after delivery. For the inclusion criteria, we selected pregnant women with a gestational age (GA) of fewer than 24 weeks who started their prenatal consultation at one of the PCUs of the municipalities and had lived in the region for at least one year before the pregnancy. Women with twin pregnancies, who had prescriptions for medications potentially neurotoxic for the fetus or pregnancy complications (or were classified as high-risk pregnancies) and women who had undergone difficult childbirths were excluded from the study. 

*Ethical issues*: This project was approved by the research ethics committee of the Faculty of Pharmacy-UFBA through Resolution 466-CNS/2012, with approval No. 3246555.

Sociodemographic Data

Trained interviewers administered semi-structured questionnaires to participants to collect data on daily habits, education level, occupation, and probable sources of PTM exposure. Socioeconomic status (SES) was stratified into five categories from A to E, based on the criteria of the Brazilian Association of Population Studies [26]. Considering the categories found (B, C, D, and E) in the study population, the SES variable was dichotomized (B/C and D/E).

### 2.2. Collection of Parameters to Assess Fetal Growth (Outcome)

FG was evaluated based on anthropometric data and GA collected from the medical records of the two maternity hospitals where the deliveries occurred. The data collected were: birth weight (BW), GA, length, and head circumference. When available, GA was primarily based on a first- or second-trimester ultrasound examination (best obstetric estimate). When ultrasound was unavailable, the estimate was based on the last menstrual period, provided by the pregnant woman. 

For the analyses, we mainly considered birth weight as the outcome variable. The other variables helped to assess the condition of the newborn. BW and GA were dichotomized as follows [27,28]: Low birth weight (LBW) if weight < 2500 g; normal weight if weight ≥ 2500 g, andPremature delivery if gestational age < 37 weeks; delivery at term if gestational age ≥ 37 weeks.

Information on the number of prenatal consultations and the baby’ sex was also collected from the medical records. The number of prenatal consultations was dichotomized into “adequate prenatal care” (≥6 visits) and “inadequate prenatal care” (<6 visits) according to the recommendations of the Ministry of Health [29]. Regarding deliveries performed in other maternity hospitals or information not found or incomplete in the medical records, we tried to recover this data from the newborn health record book during visits to the participants’ homes.

### 2.3. Assessment of Maternal Exposure to Toxic Metals (Exposure Variables)

The intrauterine exposure of newborns was estimated based on the assessment of maternal biomarkers of exposure, that is, the concentrations of the PTMs in pregnant women’s biological matrices (hair, blood, and toenails), in addition to environmental indicators (domestic settled dust). 

Sample collection and processing details are described in Bah et al. [25]. For biological samples, Pb (PbB) and Cd (CdB) were evaluated based on blood, while Mn was assessed in hair (MnH) and toenails (MnTn). Using domestic settled dust as the environmental sample, Pb levels were evaluated and expressed as the Pb dust loading rate (RtPb), µg Pb/m^2^/30 days. 

### 2.4. Conceptual Model and Hierarchy of Exposure Variables

Considering the available data and variables, we established the conceptual model for this analysis based on the DSAN-12M project model presented by Bah et al. [30]. 

The framework took a holistic approach to contaminant toxicity study in environmental science based on PTMs. Taking a multilevel perspective, the participants’ situation was divided into the four contexts of the bioecological model of Bronfenbrenner [31] (1979): (a) the microsystem—the family or proximal context for the mother-child dyad; (b) the mesosystem—the interconnection between two microsystems; (c) the exosystem—a setting in which the dyad is not directly involved but which influences its life; and (d) the macrosystem—the societal or state level. 

Figure 1 presents a schematic overview of these four contexts until delivery and shows the distribution of variables across them. In this study, the macrosystem context was little explored. Indeed, only the question about receiving government assistance (Arrow A) represents a way the executive branch could positively impact the SES of participants, as the majority are from lower social classes (D/E). However, we considered it as an exosystem variable to facilitate the analyses. At the same level, the SES and the pregnant woman’s perception of social support were also considered. In the mesosystem, prenatal visits represented opportunities for pregnant women to receive the necessary support to carry the pregnancy successfully [32]. 

We considered the dyad’s sociodemographic variables and exposure biomarkers in the microsystem. Arrows B, C and D represent how variables from other contexts influence pregnant women and the outcome in the microsystem.

This conceptual model was used as a criterion, followed by statistical aspects, to analyze the data, especially in multivariate regression. To support this, we used the statistical analysis process suggested by Leal et al. [32] and Victora et al. [33], who considered the hierarchical aspects of the studied variables. Due to limited resources, our approach was mainly limited to the microsystem level.

### 2.5. Data Analysis

Considering that the inclusion criterion was filling in the birth weight availability from hospital records, we compared the sociodemographic characteristics of the pregnant women included or not in the study population in this preliminary presentation.

A descriptive analysis of the children’s sociodemographic data, exposure, and outcome variables was presented. The Chi-squared test helped to assess possible differences. The distributions of continuous variables were evaluated using the Kolmogorov–Smirnov (KS) or Shapiro–Wilk (SW) test, and the central measures of tendency were presented as mean ± standard deviation and median (P25th–P75th). Student’s t-test was used to compare the means of birth weight according to sociodemographic variables and degrees of exposure to the PTMs.

Using Spearman’s correlation coefficient, possible relations were estimated between birth weight, other anthropometric parameters, and metal exposure biomarkers. Birth weight was used in the following ways to analyze its relationship with exposure biomarkers:Raw, orTransformed into another variable: weight (in grams)/GA (weeks) ratio, to control for the influence of GA on birth weight.

The exposure biomarker distributions, being non-parametric, were log10-transformed. Hierarchical linear regression analysis (HLR) was used to assess the association between biomarkers and FG variables (birth weight and weight/GA ratio). Following the conceptual model (Figure 1) or based on a Spearman’s correlation coefficient greater than 0.100 (*p* ≤ 0.05), the variables were included in the model.

The SPSS software version 23 for Windows was used in the statistical analysis, and a significance level of *p* < 0.05 was applied.

## 3. Results

From 164 pregnant women followed up until delivery, data on 74 births (44.5%) were collected based on completion of the birth weight data in the medical records and the baby health booklet. Depending on the variables collected in the questionnaire or medical records and the availability of biological samples, we had additional data losses.

Table 1 compares the pregnant women’s sociodemographic variables according to the inclusion in this analysis (based on FG data availability). The participants included presented a slight difference from those for whom we did not collect data. Indeed, the variables ‘waste burning’ and ‘renovated house’ showed significant differences between the two groups; lower proportions of participants who renovated their home (12.3 vs. 33.3%) or burned domestic waste (17.8 vs. 44.4%) were included. However, we found a similarity between the two groups in the exposure level to PTMs (Appendix A).

### 3.1. Mothers’ Sociodemographic Characteristics

Almost all pregnant women included in this analysis (93.1%) declared themselves as black or mixed race, with a median age of 27 ± 6.1 years and a pre-pregnancy BMI of 25.1 ± 3.8 kg/m². Nearly half (47.9%) were from low SES and lower D/E, and over two-thirds (69.2%) received government assistance. The majority (73.8%) reported being able to count on family and friends to care for their children. Almost 30% reported being passively exposed to cigarette smoke (Table 1).

### 3.2. Characteristics of Newborns and Description of Exposure Biomarkers 

Table 2 details the newborns’ characteristics and describes the exposure biomarkers. Most babies were male (55.4%), born by vaginal delivery (62.2%), and were declared as black or mixed race (89.2%). Among the pregnant women (n = 39) whose booklets noted the number of prenatal visits, almost all (92.3%) were classified as having “adequate prenatal care”.

Regarding FG parameters, only birth weight and gestational age showed a normal distribution. Considering the birth weight and GA references, few births presented a worrying situation; only 4.2 (7.7%) of the NBs were classified as LBW or were born preterm. Means ± SD for GA at delivery and birth weight were 39 ± 1.7 weeks and 3275 ± 463 g, while medians (P25th–P75th) for length and head circumference of newborns were 48 (47–50) and 34 (33–35) cm, respectively. 

Regarding exposure biomarkers, all exposure markers and biomarkers showed a non-parametric distribution by the KS test. Overall, the medians (P25th–P75th) of RtPb, PbB, MnH, MnTn, and CdB were, respectively, 13.0 (3.7–34.6) μg Pb/m²/30 days; 0.9 (0.5–1.8) µg/dL; 0.18 (0.1–0.4) µg/g; 0.65 (0.37–1.22) µg/g; and 0.54 (0.1–0.8) µg/L. Of the sociodemographic characteristics, only SES showed a significant association with MnTn (0.48 vs. 1.03 µg/g; *p* = 0.029), being higher in SES with lower D/E.

### 3.3. Influence of Sociodemographic Characteristics and Biomarkers on Birth Weight

Mothers’ sociodemographic characteristics and babies’ anthropometric data at birth (Table 3) showed no significant relationship. Birth weight and head circumference showed statistically higher means (3442 vs. 3067 g; *p* < 0.001) and medians (35 vs. 34 cm; *p* = 0.006), respectively, in boys compared to girls. 

Regarding biomarkers, bivariate analysis demonstrated no association with birth weight or BW/GA ratio in either Student’s *t*-test (Table 3) or Spearman’s coefficient correlation (Table 4). However, considering only females, MnTn showed a significant weak negative correlation (rho= −0.478; n = 24; *p* = 0.018) with the BW/GA ratio. This influence of sex can be seen in Figure 2, which shows the BW/GA ratio dispersion graphs as a function of the log of biomarkers according to sex. Although not statistically significant, LogCdB also showed a weak negative correlation (rho= −0.157; n = 31; *p* = 0.400) with the BW/GA ratio, but only for boys. 

HLR (Table 5 and Table 6) confirmed the lack of a relationship between biomarkers and outcome in the previous data presented here, although most models were statistically significant. Sex was the primary variable that maintained its association in the models. Considering a probable biological implication, we emphasize that the LogCdB showed a borderline association (beta = −2.08; *p* = 0.099) with the BW/GA ratio.

## 4. Discussion

In this study, we evaluated the impact of prenatal exposure to PTM on birth weight. In general, only Mn exposure stood out, having an inverse correlation between MnTn levels and BW/GA ratio only in girls, which suggests the effect modification of sex. Despite the worrisome exposure of the participants to Cd, we did not detect a statistically significant effect on birth weight, as suggested in other studies. The CdB showed a negative correlation only in boys, which, although statistically insignificant, also suggested a possible fetotoxicity modified by sex. It was impossible to investigate the probable combined effects of the three PTMs on birth weight due to the small sample size. Missing anthropometric measurements or other FG parameters in hospital records led to losing an essential part of the study population outcome data. 

The main inclusion criterion of this study being the availability of information on the outcome, we evaluated the possibility of extending these preliminary findings to participant groups that were not included. Despite the similarity of exposure to PTMs in the two groups, considering the importance of waste-burning and house renovation on MnTn and CdB levels [25], the lower proportion in the dyad group included in our analysis might have affected the statistical significance of the relationship between CdB and MnTn in the outcome. 

### 4.1. Exposure to PTMs

Regarding exposure to PTMs, the pregnant women had low levels of exposure to Mn and Pb, while in the case of Cd, a worrying situation was observed. Considering the references recommended in the literature [34,35], 1.9 to 3% had high levels of Mn (both MnUp and MnC), while in the case of PbB, 6.5% had levels above the CDC’s reference value (3.5 μg/dL) [36]. In the case of CdB 18.5 and 46.2% presented levels above the references for smokers (1.0 μg/L) and non-smoking Brazilian women (0.6 μg/L), respectively [37,38,39].

### 4.2. Birth Weight

The average birth weight in our study was similar to that reported by Silveira et al. [40] and Barreto et al. [41] in Brazil and Michael et al. [21] in the USA, but was greater than that reported by Goto et al. [20] in Japan. The LBW proportion was 2.3 times lower than the average prevalence found in Brazil [3]. It may be necessary to consider this proportion with caution, as we lost more than half (55.5%) of the data on FG. 

Of the sociodemographic characteristics (of the mother or the newborn), only the newborn’s sex, GA, length, and head circumference were associated with the outcome (birth weight and BW/GA ratio). This finding is not consistent with other reports in the literature; for example, cigarette smoking [12], alcohol consumption [1], receiving government assistance [42], and low SES [40] have been identified elsewhere as determinants of birth weight.

### 4.3. Relationship between Biomarkers and Outcomes

#### 4.3.1. Mn

Hair and nails are keratinized tissues and are therefore rich in cysteine and are capable of chelating metals in sulfhydryl bonds throughout their growth. These matrices showed better performance when used to investigate exposure to Mn [16,35] and to anticipate its deleterious effects in the short and long term [43]. Hair and nails, respectively, show recent (one month) and cumulative (5 to 12 months) exposure [35,43]. Few studies have evaluated Mn levels in these two matrices in pregnant women [18,44,45]. Considering the temporal window offered by toenails, MnTn provides valuable information, as it allows us to estimate the effects of Mn on FG in the first trimester of pregnancy. Unlike the second and third trimesters, some studies have considered this phase to be a time when the fetus is not subject to Mn toxicity [17,46]. Hu et al. [46] found an association between exposure to Mn and a drop in birth weight and length, while only chest circumference had a positive association in the study carried out by Mora et al. [17]. Those results are not consistent with our findings or that of Tsai et al. [47], who also found the effects of Mn on FG in the first trimester. Notably, these studies used other biological matrices (blood, hair, and urine), which could influence these differences. Like Tsai et al. [47], we investigated populations with a small sample size (n = 38 and 74, respectively), unlike Hu et al. [46] or Mora et al. [17], who examined larger populations (respectively, n = 3022 and 380). 

Another fact that must be emphasized when comparing our findings with studies carried out with other matrices was the role of sex. Although the association between MnTn and birth weight was not maintained in HLR (whether in the general population or only newborn girls), the negative correlation with birth weight only among girls is consistent with other studies that show evidence of the sex-specific toxicity of PTMs [18,48,49,50,51]. For example, in the only work (to our knowledge) that investigated the impact of Mn exposure measured by MnTn levels on FG, Signes-Pastor et al. [18] also reported a relationship affected by the sex of the newborn. The authors found a positive association for both sexes in multivariate linear regression. However, they reported an inverted-U-shaped nonlinear association only in girls. This suggests that girls are more sensitive to the effect of Mn above a certain level of exposure. This may be related to the micronutrient effect of Mn at low concentrations, while adverse effects may arise from excessive exposure. The positive association found in the linear regression might be due to the beneficial role of Mn in FG resulting from low exposure, i.e., the low level of exposure observed in this study. Indeed, this observation was consistent with the median (P25th–P75th) reported in their work: 0.32 (0.18–0.62) µg/g, two-fold lower than ours. Despite the differences observed when compared to our results, such as sample size (n = 989) and more robust statistical analyses [18], we can speculate on possible deleterious effects on FG with the median (0.65 µg/g) of our study. 

Considering only the suggested biological effects on girls (without relying on statistical significance), contrary trends can be seen from the observed correlations and the graph in Figure 2, which shows a positive correlation (rho = 0.280; n = 28; *p* = 0.149) with MnH but a negative correlation with MnTn. This difference may be due to the micronutrient status of Mn and the fact that the hair matrix indicates low recent Mn exposure, which is consistent with this tendency [10]. 

#### 4.3.2. Pb and Cd

We did not find a significant association between the endpoints and exposure to Pb and Cd in this study. However, the literature presents sufficient evidence [21,52,53,54] on toxicity of these elements to the fetus, including fetal growth. The possibility of protective factors such as diet (not evaluated so far in this report) that prevent the action of these metals may be an explanation.

In the case of Cd, we found exposure levels similar to those of workers and populations living close to industrial dump sites [55,56]. Levels similar to our findings in female participants (pregnant or not) have also been reported in other studies outside Brazil: in Canada by Garner and Levallois [57], in Greece by Sakellari et al. [58], and in the USA by Johntson et al. [53]. Considering this scenario, the limitations of this work, and the vulnerability of children during fetal life, we believe it is relevant to consider the negative association (beta = −2.08; *p* = 0.099) between LogCdB and the BW/GA ratio (although not significant) found in the HLR analysis. 

Concerning Pb, it is crucial to consider the level found in this work with caution, as it is not consistent with other studies carried out [13,59] in the district of Maragogipinho, Bahia, where we have shown high levels of exposure due to lead-glazed ceramic production. In the potters involved in this production, Bandeira et al. [60] reported a median PbB of 7.9 (0.9–49.8) µg/dL in male workers and 4.0 µg/dL in women. For this reason, our group chose to prioritize activities related to raising awareness of Pb in the region. 

Considering the outcome of this work, some studies have already demonstrated the adverse effects of Pb on birth weight at levels below the CDC reference value [6,20]. Even with a median value (P25th–P75th) of 0.63 (0.50–0.78) µg/dL, which was lower than our findings, prenatal exposure to lead was associated with decreased birth weight, increased risk of low birth weight, and small size for gestational age (birth weight < 10th percentile) [20]. 

### 4.4. Strengths and Limitations of the Study

The prospective design of this analysis and the biological plausibility of our findings meet the two main principles of causality established by Hill [59]). In the case of exposure to Mn, analyzing two matrices was an advantage, as it allowed us to consider possible consequences of exposure to Mn in the two first trimesters. Also, research such as this assessing the effects of PTMs is among the few studies carried out in populations that suffer most from health inequity in developing countries such Brazil. Besides this social aspect, the vulnerability that the pregnancy period represents for the woman and her child must be considered. 

Nevertheless, in addition to the COVID-19 pandemic, some contextual realities of the study area (participants and health center settings) hindered our data collection and may have influenced our final results, such as lack of statistical significance between PTM exposure and FG outcome. We found some resistance (participants and PCU professional teams) to participating or assisting in field data collection. Due to logistical and resource limitations, we could not locate (due to change of address or inaccessible zone) some of the pregnant women. These realities contributed to the losses observed, culminating in the small sample size. That may have led to a selection bias and the inclusion of more pregnant women with low exposure to the three PTMs because of their easier geographical accessibility, for example. Also, more than half of the medical records in the maternity hospitals or the health booklet (NB and pregnant women) were not filled out, greatly hindering data collection, and this may explain the low proportion of LBW we found. Another limitation of this study may be that it included deliveries performed by cesarean section (37.8%) and premature births, which represent a risk factor for low birth weight. Although we included them in the models as adjustments to the HLR models, their impact may persist.

## 5. Conclusions

Since pregnant women, fetuses, and children are more sensitive to PTMs, there is a need for information on their exposure and the extent to which they are affected in the southern hemisphere. Despite the limitations of this study, we detected an inverse sex-dependent relationship between low birth weight and exposure to Mn or Cd. Although we did not find any deleterious effects of Pb, it is vital to consider any level of exposure as a health risk. It is necessary to deepen this investigation, considering the population’s perception of the risk these metals pose, to reduce further exposures.

## Figures and Tables

**Figure 1 ijerph-20-06211-f001:**
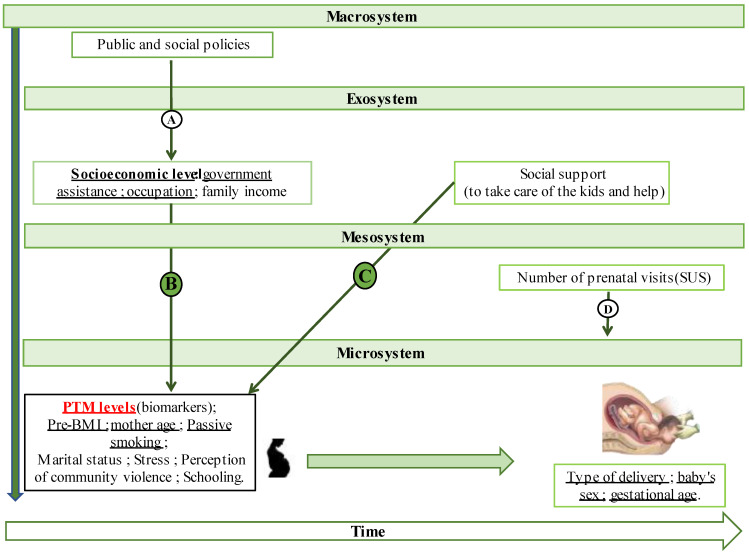
Schematic representation of the four components of the ecological context considered in our research based on the explored variables. Due to their completeness, the underlined variables were those considered in our regression model. Adapted from Victoria et al. [33] and Bah et al. [31].

**Figure 2 ijerph-20-06211-f002:**
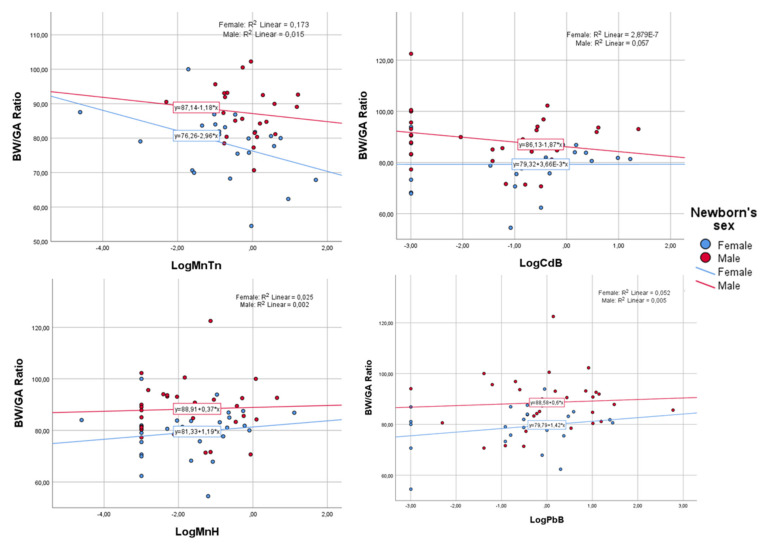
Scatter plot showing the distribution of the birth weight/GA ratio as a function of the log of biomarkers, separated by newborns’ sex.

**Table 1 ijerph-20-06211-t001:** Sociodemographic characteristics of the study population according to FG data availability (deliveries included or not in this analysis).

		FG Not Collected	FG Collected	Total	*p*-Value (*X^2^*)
Ethnicity	Black/Mixed race	86 (95.6)	68 (93.2)	154 (94.5)	0.447
	White	4 (4.4)	5 (6.8)	9 (5.5)	
Marital status	Married/stable union	53 (58.9)	36 (49.3)	89 (54.6)	0.222
	Single/divorced	37 (41.1)	37 (50.7)	74 (45.4)	
SES	D/E	42 (56.8)	35 (47.9)	77 (52.4)	0.285
	C/B	32 (43.2)	38 (52.1)	70 (47.6)	
Family income	Up to 1 salary	49 (71.0)	41 (65.1)	90 (68.2)	0.465
	Above 1 salary	20 (29.0)	22 (34.9)	42 (31.8)	
Government assistance	No	31 (44.3)	20 (30.8)	51 (37.8)	0.106
	Yes	39 (55.7)	45 (69.2)	84 (62.2)	
Education	≤Elementary school	47 (52.2)	40 (54.8)	87 (53.4)	0.743
	≥High school	43 (47.88.4)	33 (45.2)	76 (46.6)	
Occupation	Housewife	27 (30.0)	24 (32.9)	51 (31.3)	0.694
	Autonomous/other	63 (70.0)	49 (67.1)	112 (68.7)	
Passive smoker	No	60 (82.2)	52 (73.2)	112 (77.8)	0.196
	Yes	13 (17.8)	19 (26.8)	32 (22.2)	
Waste burning *	No	50 (55.6)	60 (82.29)	110 (67.5)	<0.001
	Yes	40 (44.4)	13 (17.8)	53 (32.5)	
House renovation *	No	60 (66.7)	64 (87.7)	124 (76.1)	0.002
	Yes	30 (33.3)	9 (12.3)	39 (23.9)	
Social support	No	17 (24.3)	17 (26.2)	34 (25.2)	0.803
	Yes	53 (76.1)	48 (73.8)	101 (74.8)	
		Mean ± SD	Mean ± SD	Mean ± SD	
Age (years)		27.2 ± 6.0	26.8 ± 6.3	27.0 ± 6.1	0.089
BMI pre-gestational (kg/m^2^)		25.6 ± 3.7	24.6 ± 3.9	25.1 ± 3.8	0.700

* X^2^: Chi-squared, *p* < 0.05; Student’s *t* test: *p* < 0.05. FG: fetal growth; BMI: body mass index; SD: standard deviation; SES: socioeconomic status.

**Table 2 ijerph-20-06211-t002:** Sociodemographic and anthropometric characteristics of newborns and maternal exposure markers.

Variables	Categories: n (%)			
Sex	Male	41 (55.4)	Female	33 (44.6)
Type of delivery	Vaginal	46 (62.2)	Caesarean	28 (37.8)
Antenatal visits	Inappropriate (<6 visits)	3 (7.7)	Appropriate (≥6 visits)	36 (92.3)
Low weight at birth	No (≥2500 g)	71 (95.9)	Yes (<2500 g)	3 (4.1)
Premature birth	No (≥37 weeks)	62 (92.5)	Yes (<37 weeks)	5 (7.5)
		n	Mean ± SD	Median (Q1–Q3)
Birth weight (g)		74	3275 ± 463	3290 (2955–3520)
Gestational age at delivery (Weeks)		67	39 ± 1.7	39 (38–40)
Weight/GA ratio (g/week)		67	83.9 ± 10.5	83.7 (78.5–90.5)
Length (cm)		68	48 ± 3.1	48 (47–50)
Head circumference (cm)		52	34.2 ± 1.8	34 (33–35)
RtPb (μg/m²/30 days)		36	21.8 ± 22.9	13.0 (3.7–34.6)
PbB (μg/dL)		62	1.5 ± 2.2	0.9 (0.5–1.8)
MnH (μg/g)		68	0.34 ± 0.48	0.18 (0.1–0.4)
MnTn (μg/g)		54	1.0 ± 1.02	0.65 (0.37–1.22)
CdB (μg/L)		65	0.7 ± 0.8	0.54 (0.1–0.8)

SD: standard deviation; Med (Q1–Q3): median (25th percentile–75th percentile); RtPb: Pb dust loading rate.

**Table 3 ijerph-20-06211-t003:** Birth weight comparison according to sociodemographic and exposure variables (biomarker).

	BW	*p*-Value *
	Mean ± SD	
Education		
Up to elementary school	3356 ± 362	0.113
High school or higher	3179 ± 572	
SES		
C/B	3339 ±419	0.248
D/E	3209 ±523	
Government assistance		
No	3413 ± 310	0.178
Yes	3240 ± 527	
Marital status		
Married/stable union	3301 ±397	0.675
Single/divorced	3254 ± 541	
Occupation		
Housewife	3270 ± 533	0.933
Autonomous/other	3280 ±446	
Newborn sex		
Male	3455 ± 438	<0.001
Female	3067 ± 426	
Municipality		
Nazaré	3243 ± 415	0.389
Aratuípe	3344 ± 567	
Exposure to PTMs		
PbB		
<Median	3204 ± 466	0.192
≥Median	3366 ± 497	
CdB		
<0.6 μg/L	3308 ± 544	0.713
≥0.6 μg/L	3263 ± 413	
CdB		
<1 μg/L	3270 ± 511	0.532
≥1 μg/L	3367 ± 351	
MnH		
Median	3287 ± 422	0.954
≥Median	3280 ± 542	
MnTn		
<Median	3278 ± 367	0.478
≥Median	3191 ± 498	

* Student’s *t*-test.

**Table 4 ijerph-20-06211-t004:** Spearman correlation matrix between PTM exposure markers and FG parameters.

	GA	BW/GA	Birth Weight	Length	Head Circ.
RtPb					
Rho	−0.001	−0.096	−0.005	−0.165	−0.347
P	0.997	0.535	0.979	0.359	0.083
N	32	33	35	33	26
LogMnH					
Rho	0.166	0.099	0.063	0.185	−0.184
P	0.206	0.448	0.613	0.149	0.206
N	60	61	67	62	49
LogMnTn					
Rho	0.107	−0.171	−0.077	0.022	0.005
P	0.475	0.244	0.585	0.877	0.976
N	47	48	53	50	41
LogPbB					
Rho	0.046	0.231	0.169	0.052	0.083
P	0.741	0.090	0.189	0.700	0.593
N	55	55	62	58	44
LogCdB					
Rho	0.093	−0.086	−0.055	0.092	−0.001
P	0.485	0.523	0.662	0.483	0.993
N	58	58	65	61	46

RtPb: Pb dust loading rate; Head circ.: Head circumference; Birth weight/gestational age ratio.

**Table 5 ijerph-20-06211-t005:** Summary of multivariate hierarchical linear regression (HLR) model between LogPTM and birth weight (outcome).

Models *	Beta	*p* Value	95% CI
LogPbB			
*Constant*	−3642	<0.036	−7041 to −246
*LogPbB*	−1.00	0.985	−110 to 108
*Baby sex (male)*	329	0.020	56 to 603
*Models’ statistics: n = 48; r^2^= 0.517; F = 3.497; p = 0.002*			
LogCdB ^†^			
*Constant*	−4350	0.013	−7717 to −984
*LogCdB*	−79	0.113	−177 to 20
*Baby sex (male)*	317	0.015	64 to 570
*Models’ statistics: n = 51; r2= 0.528; F = 3.454; p = 0.001*			
LogMnH			
*Constant*	−3624	0.034	−6969 to −279
*LogMnH*	28	0.630	−88 to 147
*Baby sex (male)*	269	0.035	20 to 517
*Models’ statistics: n = 53; r^2^= 0.486; F = 3.531; p = 0.002*			
LogMnTn			
*Constant*	−2769	0.064	−5607 to 168
*LogMnTn*	−40	0.523	−166 to 87
*Baby sex (male)*	293	0.011	73 to 513
*Models’ statistics: n = 40; r²= 0.686; F =5.569; p < 0.001*			

* Other predictors in the models: type of delivery (Cesarean vs. vaginal); gestational age; maternal age; pre-pregnancy BMI; SES (D/E vs. B/C); government assistance (yes vs. no); occupation (housewife vs. other). ^†^ Additional predictor in the model for the CdB: passive smoker.

**Table 6 ijerph-20-06211-t006:** Summary of multivariate hierarchical linear regression model between LogPTM and BW/GA ratio (outcome).

Models *	Beta	*p* Value	95% CI
LogPbB			
*Constant*	−17.45	0.682	−103.09 to 68.20
*LogPbB*	0.15	0.911	−2.60 to 2.90
*Baby sex (male)*	8.20	0.021	1.31 to 15.10
*Models’ statistics: n = 48; r^2^= 0.371; F = 1.928; p = 0.068*			
LogCdB ^†^			
*Constant*	−37.24	0.379	−121.97 to 47.50
*LogCdB*	−2.08	0.099	−4.58 to 0.41
*Baby sex (male)*	8.03	0.015	1.66 to 14.40
*Models’ statistics: n = 50; r^2^= 0.404; F = 2.087; p = 0.043*			
LogMnH			
*Constant*	−19.64	0.642	−104.18 to 64.92
*LogMnH*	0.68	0.648	−2.30 to 3.67
*Baby sex (male)*	6.79	0.035	0.51 to 13.07
*Models’ statistics: n = 53; r^2^= 0.348; F = 1.986; p = 0.056*			
LogMnTn			
*Constant*	−0.50	0.989	−75.95 to 74.95
*LogMnTn*	−1.15	0.478	−4.42 to 2.12
*Baby sex (male)*	7.51	0.011	1.86 to 13.16
*Models’ statistics: n = 40; r^2^= 0.579; F =3.497; p = 0.004*			

* Other predictors in the models: type of delivery (Cesarean vs. vaginal); gestational age; maternal age; pre-pregnancy BMI; SES (D/E vs. B/C); government assistance (yes vs. no); occupation (housewife vs. other). ^†^ Additional predictor in the model for the CdB: passive smoker.

## Data Availability

Not applicable.

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
