# Peer review of "Maternal Exposure to Potentially Toxic Metals and Birth Weight: Preliminary Results from the DSAN-12M Birth Cohort in the Recôncavo Baiano, Brazil"

_ijerph, 2023, doi:10.3390/ijerph20136211_

Round 1

Reviewer 1 Report

Prenatal exposure to potentially toxic metals (PTMs), such as cadmium, lead, and manganese, may impair fetal growth resulting in low birthweight. There is a growing concern of the consequences of multiple exposures to PTMs, and the studies conducted in low and middle-income countries are needed. Bah et al. assessed the association of PTM levels and birthweight in DSAN-12M birth cohort in Brazil, as well as the influences of infant sex and their socioeconomic levels. I have some suggestions and points of clarification below.

1.     The authors mentioned in Line 313 that the LBW proportion in current study was 2.3 times lower than the average in the country. Are there any explanations for this observation? or possibly due to small sample size? Does this contribute to the findings that most PTM levels were not associated with low birth weight as suggested by many other published studies? Please discuss.

2.     Were there any criteria for pregnancy complications, such as gestational diabetes, in current study? If so, please write it clearly in the Method section. If women with pregnancy complications were not excluded from the study, the number and types of complications should be presented in the results and discussed, since women with gestational diabetes will tend to have bigger babies that may affect the correlation analysis.

3.     Were the information of diet/nutrients/food security, medicine or vitamins used collected in the study? All these factors may affect the birth weight as well as toxicity of PTM.

4.     Since the levels of PTM didn’t affect birthweight significantly in the current study, other parameters indicating poor growth/development associated with PTM exposure may be assessed. please discuss briefly.

5.     Figure 1 is unnecessary, please remove it.

6.     Figure 3 is redundant. This can be indicated clearly in the text or figure 2.

7.     In Line 122-123, the definitions of adequate/inadequate prenatal care are opposite of Table 2.

8.     Figure 4 should be removed. Similar diagram has been published in Int. J. Environ. Res. Public Health 2023, 20, 2949. Just need to cite the paper and state the number of cases with FG data that was used in the current study.

9.     In Table 1, what is IMC? or it should be BMI?

Author Response

Response to Reviewer Comments

Prenatal exposure to potentially toxic metals (PTMs), such as cadmium, lead, and manganese, may impair fetal growth resulting in low birthweight. There is a growing concern of the consequences of multiple exposures to PTMs, and the studies conducted in low and middle-income countries are needed. Bah et al. assessed the association of PTM levels and birthweight in DSAN-12M birth cohort in Brazil, as well as the influences of infant sex and their socioeconomic levels. I have some suggestions and points of clarification below.

Point 1: The authors mentioned in Line 313 that the LBW proportion in current study was 2.3 times lower than the average in the country. Are there any explanations for this observation? or possibly due to small sample size? Does this contribute to the findings that most PTM levels were not associated with low birth weight as suggested by many other published studies? Please discuss.

Response 1: Thank you for the suggestion; we have considered your recommendation. We reviewed the section “4.4. Strengths and limitations of the study” and rediscussed it accordingly.

Point 2: Were there any criteria for pregnancy complications, such as gestational diabetes, in current study? If so, please write it clearly in the Method section. If women with pregnancy complications were not excluded from the study, the number and types of complications should be presented in the results and discussed, since women with gestational diabetes will tend to have bigger babies that may affect the correlation analysis.

Response 2: Thank you for your comment. No; participants with pregnancy complications were not included in the study as those situations may influence newborn birth weight. We have considered your recommendation and report this information in the “2.1. Study population” section.

Point 3: Were the information of diet/nutrients/food security, medicine or vitamins used collected in the study? All these factors may affect the birth weight as well as toxicity of PTM.

Response 3:  Thank you for your observation. Yes, data about medicine or vitamins have been collected. Some (n=2) were taking medicine with neurological effects and were not included in the study sample. Almost all participants of our population sample (n= 73 or 98,6%) were taking iron and vitamins get at the primary care units (PCU) of the Brazilian Public Health System (SUS is its acronym in Portuguese).

Unfortunately, food insecurity was not assessed in our study, and we applied a questionnaire on diet (Food Frequency Questionnaire) only to half (n=35) of the population study included in this analysis. We have included a sentence in our discussion about this limit of our paper: "The possibility of protective factors such as diet (not evaluated so far in this report) that prevent the action of these metals may be an explanation."

Point 4: Since the levels of PTM didn’t affect birth weight significantly in the current study, other parameters indicating poor growth/development associated with PTM exposure may be assessed. please discuss briefly.

Response 4: Thank you for your comment. Actually, we have also assessed relationship between PTMs and gestational age or others anthropometric parameters. However, the results were inconclusive; probably due to the worse quality of those data (when compared with birth weight).

Point 5: Figure 1 is unnecessary, please remove it.

Response 5: Thank you for your suggestion. The figure 1 has been removed as suggested.

Point 6: Figure 3 is redundant. This can be indicated clearly in the text or figure 2.

Response 6: Thank you for your suggestion. The figure 3 has been removed and the idea transferred to figure 1 (which was the figure 2)

Point 7: In Line 122-123, the definitions of adequate/inadequate prenatal care are opposite of Table 2.

Response 7: Thank you for observing this mistake. We made the correction accordingly.

Point 8: Figure 4 should be removed. Similar diagram has been published in Int. J. Environ. Res. Public Health 2023, 20, 2949. Just need to cite the paper and state the number of cases with FG data that was used in the current study.

Response 8: Thank you for your suggestion. We have considered it and remove the figure 4.

Point 9: In Table 1, what is IMC? or it should be BMI?

Response 9: Thank you for observing this mistake. We made the correction accordingly.

Reviewer 2 Report

Ferréol Bah et al have carried out a clinical study in which they have tried to verify whether exposure to certain metals during pregnancy translates into low birth weight in newborns. Although the sample size is not very high, the study suggests a certain influence of manganese and cadmium on the studied effect.

The work is correctly written, includes an introduction section sufficient to present the topic, the methodology is clearly and in detail described, the results are correctly presented and discussed; and the conclusion is consistent with them.

I have not detected any important aspect that should be reviewed or corrected. Just a few minor observations that should be resolved before definitively publishing the work:

- Line 24: what the abbreviation GA means has not been described in the abstract.

- Figure 4 has not been presented in the text.

- Abbreviations used in tables and figures should be described in the table or figure captions.

- Table 1: "IMC" in English is abbreviated as BMI (body mass index).

- Table 2: the correct unit to express the mass is "g", not "grams".

Author Response

Response to Reviewer Comments

Ferréol Bah et al have carried out a clinical study in which they have tried to verify whether exposure to certain metals during pregnancy translates into low birth weight in newborns. Although the sample size is not very high, the study suggests a certain influence of manganese and cadmium on the studied effect.

The work is correctly written, includes an introduction section sufficient to present the topic, the methodology is clearly and in detail described, the results are correctly presented and discussed; and the conclusion is consistent with them.

I have not detected any important aspect that should be reviewed or corrected. Just a few minor observations that should be resolved before definitively publishing the work:

-

Point 1: Line 24: what the abbreviation GA means has not been described in the abstract.

Response 1: Thank you for the comment; GA means Gestational Age. We made the correction accordingly.

Point 2: Figure 4 has not been presented in the text.

Response 2: Thank you for your comment. Based on suggestion of another reviewer, we have removed the figure 4.

Point 3: Abbreviations used in tables and figures should be described in the table or figure captions.

Response 3: Thank you for your observation. We made the correction accordingly.

Point 4: Table 1: "IMC" in English is abbreviated as BMI (body mass index);

Table 2: the correct unit to express the mass is "g", not "grams".

Response 4: Thank you for observing this mistake. We made the correction accordingly.

Reviewer 3 Report

Dear authors,

The study investigate the association between exposure to potentially toxic metals such as lead, cadmium and manganese to children’s birth weight and possibility of a combined influence of these metals on birthweight, taking into account the social context of the study population. The work shows some trends, but without statistically significant effects. The paper present interesting data, but authors should avoid repeating the early part of their published data.

Particular comments below:

 L24: GA – the contraction is used the first time and should be explain.

L90: –  “gestational age” this is the first time used in the text, and the abbreviation should be placed here. 

L132-133: “Details of biological sample collection and processing are described in Bah et al. [25]. Briefly, Pb levels were evaluated in domestic settled dust expressed in lead dust deposition rate μg Pb/m2/30 days. “

The first sentence is about collection and processing of biological samples, but next sentence should contain its briefly description of this process, but contain information about domestic dust. In my opinion this paragraph is incoherent and should be improved.

L215: FG - should be explained also in the text

L247: Why were means used for birth weight and medians for head circumference?

L421: Pregnant women, fetuses, and children are more sensitive to PTMs.

This is true, but it is not the conclusion from this work.

L424: “This work investigated possible relationships between three PTMs and birthweight, a parameter of fetal growth.” It is not a conclusion.

Author Response

Response to Reviewer Comments

Dear authors,

The study investigates the association between exposure to potentially toxic metals such as lead, cadmium and manganese to children’s birth weight and possibility of a combined influence of these metals on birthweight, taking into account the social context of the study population. The work shows some trends, but without statistically significant effects. The paper present interesting data, but authors should avoid repeating the early part of their published data.

Particular comments below:

Dear reviewer, thank you very much for all comments and suggestion. Dully noted.

Point 1: L24: GA – the contraction is used the first time and should be explain.

Response 1: Thank you for the comment; GA means Gestational Age. We made the correction accordingly.

Point 2: L90: “gestational age” this is the first time used in the text, and the abbreviation should be placed here.

Response 2: Thank you for observing this mistake. We made the correction accordingly.

Point 3: L132-133: “Details of biological sample collection and processing are described in Bah et al. [25]. Briefly, Pb levels were evaluated in domestic settled dust expressed in lead dust deposition rate μg Pb/m2/30 days“. The first sentence is about collection and processing of biological samples, but next sentence should contain its briefly description of this process, but contain information about domestic dust. In my opinion this paragraph is incoherent and should be improved.

Response 3: Thank you for your observation. We reformulated this paragraph to let it more coherent. Please read now: “Sample collection and processing details are described in Bah et al. [25]. In the case of biological samples, Pb (PbB) and Cd (CdB) were evaluated in blood, while Mn was assessed in hair (MnH) and toenails (MnTn). With domestic settled dust as the environmental sample, Pb levels were evaluated and expressed as deposition rate µg Pb/m2/30 days.”

Point 4: L215: FG - should be explained also in the text

Response 4: Thank you for the suggestion; we have considered your recommendation and revising the whole manuscript where Fetal Growth has been used.

Point 5: L247: Why were means used for birth weight and medians for head circumference?

Response 5: Thank you for your question. Birth weight presented a normal distribution while head circumference was non parametric. That is why we presented the results differently.

Point 6: L421: Pregnant women, fetuses, and children are more sensitive to PTMs. This is true, but it is not the conclusion from this work.

Response 6: Thank you for your comment. We have considered your recommendation and reviewed the whole conclusion.

Point 7: L424: “This work investigated possible relationships between three PTMs and birthweight, a parameter of fetal growth.” It is not a conclusion.

Response 7: Thank you for observing this mistake. We removed this part from the conclusion.

Reviewer 4 Report

1. Authors must consider the BMI and gestational diabetes/ preeclampsia incidence in the mothers

2. The tables are confusing and must be presented more clearly.

3. Authors must limit the discussion sections to the main findings of the study instead of a literature review-type approach

Author Response

Response to Reviewer Comments

Point 1: Authors must consider the BMI and gestational diabetes/preeclampsia incidence in the mothers

Response 1: Thank you for the suggestion; we have considered your recommendation. Yes, participants with pregnancy complications were not included in the study as those situations may influence newborn birth weight. We have considered your recommendation and include this information in the “2.1. Study population” section.

Point 2: The tables are confusing and must be presented more clearly.

Response 2: Thank you for your comment. We have considered your recommendation and adjusted to let them clearer.

Point 3: Authors must limit the discussion sections to the main findings of the study instead of a literature review-type approach

Response 3: Thank you for your observation. We have considered your recommendation and reviewed the discussion section accordingly.

Reviewer 5 Report

This is an elegant study investigating the role of exposure to potentially toxic metals in pregnant mothers that may impair fetal growth. The methods are sound, and results and their interpretation are very clear. A few comments are provided that may help to clear up few small details for the readers.

-The figures need to have consistent font size as well as either use comma (,) or the semicolon (;) to keep it consistent.

-Figure 4 flowchart has a comma in 54,9% instead of 54.9% (FG data not collected box)

-All table legends need to describe the statistical method used for analysis.

-Line 238, does not define what RtPb is.

Author Response

Response to Reviewer Comments

 This is an elegant study investigating the role of exposure to potentially toxic metals in pregnant mothers that may impair fetal growth. The methods are sound, and results and their interpretation are very clear. A few comments are provided that may help to clear up few small details for the readers.

Point 1: -The figures need to have consistent font size as well as either ruse comma (,) or the semicolon (;) to keep it consistent.

Response 1: Thank you for the suggestion; we have considered your recommendation and reviewed the figures.

Point 2: Figure 4 flowchart has a comma in 54,9% instead of 54.9% (FG data not collected box)

Response 2: Thank you for your comment. Based on suggestion of another reviewer, we have removed the figure 4.

Point 3: All table legends need to describe the statistical method used for analysis.

Response 3: Thank you for your observation. We added this information as you suggested.

Point 4: Line 238, does not define what RtPb is.

Response 4: Thank you for the comment; RtPb means Lead Dust Loading Rate. We have considered your recommendation and inform the use of this abbreviation at the first appearance of this expression. We marked in yellow the parts corrected in the document.

Reviewer 6 Report

The study addresses an important issue in maternal health.

General Comments: The paper is well-written and addresses an important research question. However, the authors need to address a few issues to enable it to appeal to a wider international audience:

Specific Comments:

Ø  The abstract is adequate in form and content. It has included all the aspects of the work. Just include a line on the relevance or implications of the findings from these preliminary results.

Ø  Contextualize the statement in 59.

Ø  Delete 'Briefly' in line 81.

Ø  Delete the word 'thorough' in line 83.

Ø  On Page 3, lines 98-99 can be reviewed for clarity and coherence. And let it be an active sentence.

Ø  Lastly, make sure all the citations are references accordingly.

Author Response

Response to Reviewer Comments

REVIEWER 6

The study addresses an important issue in maternal health.

General Comments:

The paper is well-written and addresses an important research question. However, the authors need to address a few issues to enable it to appeal to a wider international audience: 

Point 1: The abstract is adequate in form and content. It has included all the aspects of the work. Just include a line on the relevance or implications of the findings from these preliminary results.

Response 1: Thank you for the suggestion; we have considered your recommendation and included a sentence accordingly. Please read: These preliminary results confirmed the threat PTMs may represent and the need for more extensive and further research to elucidate their role in inadequate FG in developing countries.”

Point 2: Contextualize the statement in 59.

Response 2: Thank you for your comment. we have considered your recommendation and revised the sentence accordingly. Please, now read: “In real life, humans are commonly exposed to different types of pollutants simultaneously. There is a growing concern about the consequences of multiple exposures to PTMs.”

Point 3: Delete 'Briefly' in line 81.

Response 3: Thank you for your observation, we have considered your recommendation.

Point 4: Delete the word 'thorough' in line 83.

Response 4: Thank you for the suggestion; we have considered your recommendation.

Point 5: On Page 3, lines 98-99 can be reviewed for clarity and coherence. And let it be an active sentence.

Response 5: Thank you. We have considered your recommendation and include a sentence accordingly. Please, now read: “Trained interviewers applied semi-structured questionnaires to participants to collect data on daily habits, education level, occupation, and probable sources of PTM exposure”

Point 6: Lastly, make sure all the citations are references accordingly.

Response 6: Thank you for your suggestion. We have checked the citations in the reference list.

Round 2

Reviewer 4 Report

No comments